# The Depression, Anxiety, Stress Scales-21: Principal component analysis and evaluation of abbreviated versions in young adults with temporomandibular disorders

Yinghao Xiong[1], Adrian Ujin Yap[1,2,3,4], Carolina Marpaung[4], May Chun Mei Wong[1]*

1 Faculty of Dentistry, The University of Hong Kong, Hong Kong, Hong Kong, 2 Department of Dentistry, Ng Teng Fong General Hospital and Faculty of Dentistry, National University Health System, Singapore, Singapore, 3 National Dental Research Institute Singapore, National Dental Centre Singapore and Duke-NUS Medical School, Singapore, Singapore, 4 Department of Prosthodontics, Faculty of Dentistry, Universitas Trisakti, Jakarta, Indonesia

* mcmwong@hku.hk

**Data Availability Statement:** All relevant data are within the paper and its Supporting information files.

## Abstract

### Background

The Depression, Anxiety, Stress Scales-21 (DASS-21) contain three subscales measuring depression, anxiety, and stress. Several abbreviated DASS-21 versions have been developed, demonstrating better clinical utility and measurement properties than the original instrument. This study explored the factor structure of various abbreviated DASS-21 versions and identified/validated the optimal one for assessing young adults with temporomandibular disorders (TMDs).

### Methods

A total of 974 university-attending young adults were recruited in two waves (wave 1: 519; wave 2: 455). Demographic information, the DASS-21, and quintessence five TMD symptoms (5Ts) of the Diagnostic Criteria for TMDs were collected. Principal component analysis (PCA) was employed to condense the DASS-21 (wave 1 data), while confirmatory factor analysis (CFA) was used to determine maximum likelihood estimates and compare different abbreviated DASS-21 versions (wave 2 data). Known-group, concurrent (criterion) validity and reliability were subsequently evaluated.

### Results

The mean age of the study participants was 21 (SD = 0.1) years and 80.4% were women. Twelve DASS-21 items were identified from the PCA. However, the Korean DASS-12 provided the best-fit model ($\chi^2$/df = 2.07, CFI = 0.975, TLI = 0.960, RMSEA = 0.049, SRMR = 0.033) among the seven abbreviated versions in the CFA. The Korean DASS-12 showed good known-group and concurrent ($r_s$ = 0.959) validity and reliability when contrasted to the DASS-21.

**Funding:** The author(s) received no specific funding for this work.

**Competing interests:** The authors have declared that no competing interests exist.

**Abbreviations:** 3Q/TMD, Three screening questions; 5Ts, The quintessential five temporomandibular disorder symptoms; AUCs, Receiver operating characteristic curves; BDI/BAI, Beck Depression/Anxiety Inventory; CFA, Confirmatory factor analysis; CFI, Comparative fit index; DASS, Depression Anxiety Stress Scale; DC/TMD, Diagnostic Criteria for TMDs; EFA, Exploratory factor analysis; GHQ, General Health Questionnaire; HADS, Hospital Anxiety and Depression Scale; IT, Intra-articular; KMO, Kaiser-Meyer-Olkin; MAP, Minimum average partial; PA, Parallel analysis; PCA, Principal component analysis; PHQ, Patient Health Questionnaire; PT, Pain-related; RMSEA, Root mean square error of approximation; SFAI, The Short-form Fonseca Anamnestic Index; SRMR, Standardised root mean square residual; TLI, Tucker Lewis index; TMDs, Temporomandibular disorders; TMJs, Temporomandibular joints; TPS, TMD Pain Screener.

## Conclusion

The Korean DASS-12 possessed a good fit, known-group, as well as concurrent (criterion) validity and reliability, and was the best abbreviated DASS-21 version for screening young adults with TMD symptoms for psychological distress.

## Background

Depression and anxiety are two prevalent negative effects experienced by individuals with physical and mental disorders [1]. Characterized by persistent feelings of sadness and a lack of interest, depression often goes hand-in-hand with anxiety, an emotion marked by tension and worrisome thoughts [2]. To assess these mental health challenges, numerous reliable and valid scales have been developed. Some of the most prominent include the Hospital Anxiety and Depression Scale (HADS), Beck Depression/Anxiety Inventory (BDI/BAI), General Health Questionnaire (GHQ), Patient Health Questionnaire (PHQ), and Depression Anxiety Stress Scale (DASS) [3]. The 21-item condensed version of the DASS, known as DASS-21, has gained popularity due to its ability to simultaneously evaluate three negative emotional constructs: depression, anxiety, and stress. DASS-21 has been translated into different languages and validated in different populations [4–7]. As a reliable, valid, and accurate tool, the DASS-21 compares favorably to the original DASS-42 and has been translated into multiple languages for use in both clinical and non-clinical assessments [1, 8].

DASS-21 contains three subscales with seven items each for assessing the emotional states. Subscale scores are calculated by summing the item scores, and each subscale has distinct cut-off points for severity ratings, ranging from normal to extremely severe [1]. Shorter DASS versions with fewer items can increase research participation and data quality, particularly in epidemiological studies [9]. A number of abbreviated versions of the DASS-21 (Table 1) have been developed that claim better measurement properties than the DASS-21. These include the DASS-18 [10], DASS-14 [11], Malaysian DASS-12 [12], Korean DASS-12 [13] and DASS-8 [14]. Additionally, the factor structure and dimensionality of DASS-21 have also been questioned recently. A comprehensive systematic review [8] indicated sufficient high-quality evidence to support the bifactor structure of DASS-21 (Fig 1). Besides, Zanon et al. [15] examined the dimensionality, reliability, and invariance across eight countries and supported the use of a general factor of distress rather than three factors. Furthermore, Yap and Lee [16] found that the DASS-21 only contained two factors instead of the three initially stated. Therefore, the dimensionality of DASS-21 remains uncertain and could vary depending on the study population.

Temporomandibular disorders (TMDs) are a diverse group of conditions involving pain and dysfunction of the temporomandibular joints (TMJs), masticatory muscles, and related structures [17, 18]. Up to 15% of adults and 7% of adolescents are affected by TMDs, and chronic pain is the primary reason for treatment-seeking [19]. Females have a more than two-fold likelihood of experience TMD than males [20]. According to the contemporary Diagnostic Criteria for TMDs (DC/TMD) standard, TMDs can be divided into pain-related (PT) and intra-articular (IT) problems [21]. Several screening instruments have been developed to detect the presence of TMD [22], including the TMD Pain Screener (TPS), three screening questions (3Q/TMD), the Short-form Fonseca Anamnestic Index (SFAI), and the quintessential five temporomandibular disorder symptoms (5Ts). The TPS is part of the DC/TMD repertoire but is specifically designed for assessing the presence of painful TMDs [23]. Despite

**Table 1. Depression Anxiety Stress Scale and the abbreviated versions.**

| Items | Questions | Subscales | DASS-21 | DASS-18 | DASS-14 | Malaysian DASS-12 | Korean DASS-12 | DASS-8 |
|---|---|---|---|---|---|---|---|---|
| I1 | I found it hard to wind down | S | ✓ | ✓ | ✓ | ✓ | ✓ | |
| I2 | I was aware of dryness of my mouth | A | ✓ | ✓ | | ✓ | ✓ | |
| I3 | I couldn't seem to experience any positive feeling at all | D | ✓ | ✓ | ✓ | ✓ | ✓ | |
| I4 | I experienced breathing difficulty (e.g., excessively rapid breathing, breathlessness in the absence of physical exertion) | A | ✓ | ✓ | ✓ | ✓ | ✓ | |
| I5 | I found it difficult to work up the initiative to do things | D | ✓ | ✓ | | | | |
| I6 | I tended to over-react to situations | S | ✓ | ✓ | ✓ | ✓ | | |
| I7 | I experienced trembling (e.g., in the hands) | A | ✓ | ✓ | ✓ | ✓ | ✓ | |
| I8 | I felt that I was using a lot of nervous energy | S | ✓ | | | | | ✓ |
| I9 | I was worried about situations in which I might panic and make a fool of myself | A | ✓ | ✓ | | | | ✓ |
| I10 | I felt that I had nothing to look forward to | D | ✓ | ✓ | ✓ | ✓ | ✓ | ✓ |
| I11 | I found myself getting agitated | S | ✓ | | ✓ | | ✓ | |
| I12 | I found it difficult to relax | S | ✓ | | ✓ | | ✓ | ✓ |
| I13 | I felt down-hearted and blue | D | ✓ | ✓ | | | ✓ | ✓ |
| I14 | I was intolerant of anything that kept me from getting on with what I was doing | S | ✓ | ✓ | ✓ | ✓ | | |
| I15 | I felt I was close to panic | A | ✓ | ✓ | | | | ✓ |
| I16 | I was unable to become enthusiastic about anything | D | ✓ | ✓ | ✓ | | | ✓ |
| I17 | I felt I wasn't worth much as a person | D | ✓ | ✓ | ✓ | ✓ | ✓ | |
| I18 | I felt that I was rather touchy | S | ✓ | ✓ | ✓ | ✓ | ✓ | |
| I19 | I was aware of the action of my heart in the absence of physical exertion (e.g., sense of heart rate increase, heart missing a beat) | A | ✓ | ✓ | ✓ | ✓ | ✓ | |
| I20 | I felt scared without any good reason | A | ✓ | ✓ | | | | ✓ |
| I21 | I felt that life was meaningless | D | ✓ | ✓ | ✓ | ✓ | | |

detecting both pain-related (PT) and intra-articular (IT) TMDs, the three screening questions (3Q/TMD) have limited accuracy in identifying individuals who met the PT and IT TMD criteria according to the DC/TMD standard. Specifically, only 74% of individuals who tested positive and 16% of those who tested negative on the 3Q/TMD met the PT and IT TMD criteria [24]. The SFAI comprises five items and is an abbreviated version of the Fonseca Anamnestic Index. The SFAI presents high accuracy in detecting TMD when referenced to the DC/TMD [17]. The 5Ts questionnaire, founded on the DC/TMD Symptom Questionnaire (SQ), involves the quintessential five TMD symptoms: TMD pain, headache, TMJ noises, closed and open locking [21]. It also demonstrated high accuracy in identifying the presence of TMDs when referenced to the DC/TMD benchmark. When compared to the SFAI, the 5Ts had higher sensitivity and specificity for detecting pain-related or intra-articular TMDs [17, 24].

The multifactorial aetiology of TMD includes biological, psychological, and social factors. TMD symptoms, especially pain, are considered a cause of psychological distress [25, 26], and TMD patients were found to have higher levels of psychological distress than people without TMDs [19]. The DC/TMD criteria advocate for the utilization of the PHQ-4, PHQ-9, and Generalized Anxiety Disorder-7 (GAD-7) instruments to measure distress, depression, and anxiety individually. These assessment tools enable a more targeted evaluation of each specific mental health concern, providing valuable insights for diagnosis and treatment [21]. The DASS-21 could be a more comprehensive tool for screening TMD patients from three domains. However, the reliability and dimensionality of the DASS-21 for use in individuals

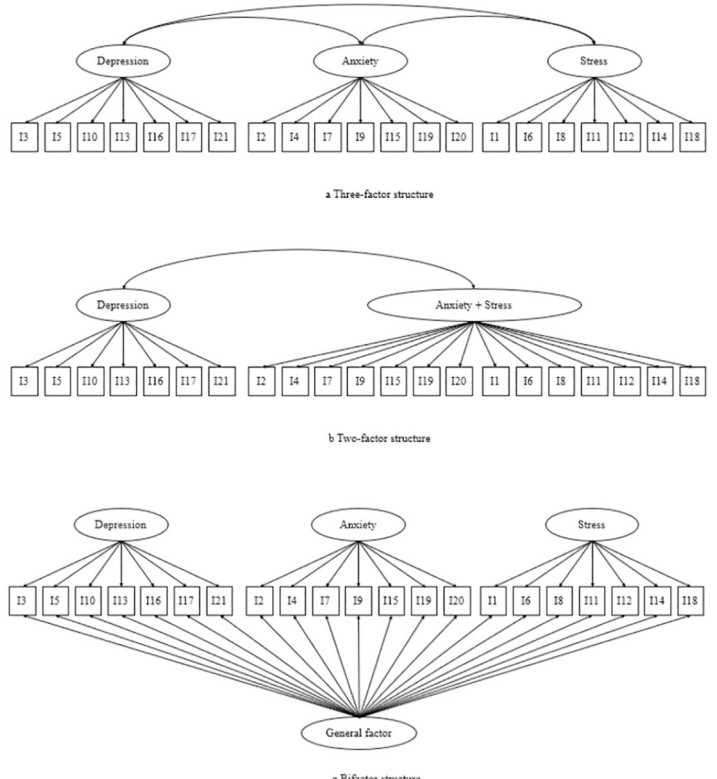

**Fig 1. Three-factor, two-factor and bifactor structures.**

with TMDs had been queried, and subscale discrepancies were observed [16]. Though the DASS-21 was designed for clinical and non-clinical samples, its validity in measuring psychological distress, specifically in individuals with TMDs, has not been extensively studied. Since different clinical conditions often manifest varying psychological comorbidities, symptom profiles, and severity, it is essential to examine the underlying factor structure of the DASS-21 and its abbreviated versions in individuals with TMDs to increase participation and data quality in both clinical and research settings.

This study aimed to (a) shorten the DASS-21 using principal component analysis (PCA), (b) confirm the factor structure and model fit of various abbreviated DASS-21 versions using confirmatory factor analysis (CFA), and (c) identify and establish the validity/reliability of the optimal version for assessing young adults with TMDs in Indonesian young adults.

## Methods

Ethics approval for this work was obtained from the institution review board of the Faculty of Dentistry, Universitas Trisakti, Jakarta, Indonesia (ID: 377/S1/KEPK/FKG/8/2020 and 017/S3/KEPK/FKG/12/2021). Data were acquired in two waves, with the first (n = 519) and second (n = 455) waves collected from January to May and June to October 2021, respectively. Participants were recruited from a major private university in Jakarta, the capital city, through a convenience sampling technique involving intranet posting and face-to-face engagements. The inclusion criteria were young adults aged 18 to 25 with good general health. The exclusion criteria were individuals with a history of traumatic injuries and those with debilitating psychological and physical (metabolic, autoimmune, and other systemic problems). Those who

presented incomplete questionnaires were also duly omitted. All the study participants provided informed consent and voluntarily completed an online questionnaire (Google Forms) encompassing demographic information, the DASS-21, and 5Ts.

The minimum sample size was determined by the number of participants required for conducting factor analysis. Recommendations of sample sizes for factor analysis are as follows: poor/fair– 100 to 200 participants; reasonable– 300 participants; very good– 500 participants [27]. Regarding the participants-to-variable ratios, at least five participants per measured variable have been proposed, with a minimum of 100 participants [28]. Alternatively, another recommendation was 10:1 or 20:1 in terms of the ratio between participants and measured variables [29]. Considering the above criteria and adopting the ratio of 20:1, at least 420 participants were needed for the study.

Information collected and instruments used in this study included:

*Demographic information*: The demographic information collected consisted of age and gender.

*The quintessential five temporomandibular disorder symptoms (5Ts)*: The 5Ts was used to establish the presence of TMD symptoms. It had high accuracy for detecting PT, IT, and all TMDs with the area under the receiver operating characteristic curves (AUCs) of 1.00, 0.98, and 0.98, respectively. The diagnostic performance was good, with specificity of 100% and sensitivity of 96.1% to 99.2% [17]. Participants were considered 5Ts-positive if they answered "yes" to any of the five questions. If they replied "no" to all five questions, they were 5Ts-negative and considered without TMD symptoms.

*Depression Anxiety Stress Scales-21*: The English version has been used in the study [1]. DASS-21 contains 21 items measuring three distinct negative effects: Depression (items 3, 5, 10, 13, 16, 17, 21), Anxiety (items 2, 4, 7, 9, 15, 19, 20) and Stress (items 1, 6, 8, 11, 12, 14, 18). Items are scored on a 4-point scale from 0 ("did not apply to me at all") to 3 ("applied to me very much or most of the time"). Greater scores for each subscale (sum of the item scores) suggested higher levels of psychological distress, and cut-off points for different severity groupings are reflected in the DASS manual [1].

Data were analyzed using IBM SPSS Statistic v28.0.1.0 (IBM Corporation, Armonk, NY, USA) and R Studio v4.1.3 (RStudio, PBC, Boston, MA, USA) with the significance level set at 0.05 where applicable. Data normality was examined through multivariate normality test using Mardia coefficients, skewness and kurtosis, with skewness $> 2.0$ or kurtosis $> 7.0$ indicating severe nonnormality [30], Bartlett's test of sphericity and Kaiser-Meyer-Olkin (KMO) test of sampling adequacy were performed. Bartlett's test showed a significant result ($p<0.05$), and a KMO value $\geq 0.8$ indicated that the data were adequately sampled and appropriate for analysis.

PCA was used to shorten the DASS-21 using wave 1 data. A combination of the parallel analysis (PA) method [31], minimum average partial (MAP) method [32], and scree plot was used to identify the number of components to be retained [33]. Spearman correlation was used because the data distribution had low kurtosis and the sample size was not small [34]. Oblique rotation was chosen because these subscales were correlated, and among all the possible analytic processes, Promax was selected as a widely accepted modification of the varimax procedure [35]. Based on the practical and statistical significance consideration [36], the meaningful threshold for factor loadings was set at 0.6 in this study.

CFA using maximum likelihood estimation was used to compare different versions (DASS-21, DASS-18, DASS-14, the Malaysian DASS-12, the Korean DASS-12, the current DASS-12 and DASS-8) using wave 2 data. Three-factor structure (Depression, Anxiety, and Stress), two-factor structure (Depression and Anxiety-Stress), and bifactor structure (Depression, Anxiety, Stress, and a general factor) were assessed (Fig 1). However, identification problems exist

when applying the bifactor model to factors containing two items. Two methods were applied to solve this problem (S1 Appendix) [37]. Various fit indices were used to confirm the model fit. Data is considered to have a good fit, if the Comparative fit index (CFI) is > 0.90, Tucker Lewis index (TLI) is > 0.90, the root mean square error of approximation (RMSEA) is < 0.06, and the standardised root mean square residual (SRMR) is < 0.06 [37].

Known-group validity was examined by comparing the Depression, Anxiety, and Stress scores between young adults with and without TMD symptoms (excluding individuals with only headaches) using the Mann-Whitney test in SPSS with the significance level set at 0.05. Data normality was tested using the Shapiro-Wilk test. Concurrent (criterion) validity was examined using Spearman correlation. The reliability was assessed through internal consistency of the total and subscale scores of the DASS-21 and the optimal abbreviated version utilizing Cronbach's Alpha. Cronbach's Alpha higher than 0.7 was regarded as acceptable [38].

## Results

A total of 974 university-attending young adults were recruited in two waves. Wave 1 (n = 519) comprised 393 females (75.7%) and 126 males (24.3%), with a mean age of 19.8 (SD = 1.3) years. Wave 2 (n = 455) comprised 390 females (85.7%) and 65 (14.3%) males, with a mean age of 22.4 (SD = 1.3) years.

Descriptive statistics (Table 2) of individual items were examined. Multivariate normality test using Mardia coefficients considered the data as multivariate non-normality distribution (Mardia skewness statistic = 53.72, p<0.001; Mardia kurtosis statistic = 578.16, p<0.001). All measured variables were considered not severely deviated from normal distribution (skewness < 2 and Kurtosis < 4). Bartlett's test of sphericity (chi-square of 4122.9 with 210 degrees of freedom, p < 0.001) and the values of the KMO test of sampling adequacy were 0.93

**Table 2. Descriptive statistics and KMO results (Wave 1 data, n = 519).**

| Items | Mean (SD) | Skewness | Kurtosis | KMO value |
|---|---|---|---|---|
| I1 | 1.1 (0.8) | 0.49 | 0.02 | 0.95 |
| I2 | 0.8 (0.7) | 0.80 | 0.59 | 0.90 |
| I3 | 0.7 (0.7) | 0.97 | 0.85 | 0.93 |
| I4 | 0.6 (0.8) | 1.14 | 0.65 | 0.89 |
| I5 | 0.9 (0.8) | 0.67 | 0.18 | 0.86 |
| I6 | 1.1 (0.9) | 0.49 | -0.35 | 0.93 |
| I7 | 0.8 (0.8) | 0.83 | 0.03 | 0.91 |
| I8 | 1.2 (0.9) | 0.34 | -0.79 | 0.94 |
| I9 | 1.4 (1.0) | 0.09 | -0.96 | 0.95 |
| I10 | 0.4 (0.7) | 1.81 | 3.01 | 0.93 |
| I11 | 1.1 (0.8) | 0.55 | -0.11 | 0.94 |
| I12 | 1.0 (0.8) | 0.67 | 0.26 | 0.94 |
| I13 | 1.1 (0.9) | 0.64 | -0.17 | 0.94 |
| I14 | 1.0 (0.8) | 0.59 | -0.20 | 0.87 |
| I15 | 1.4 (1.0) | 0.23 | -0.94 | 0.94 |
| I16 | 0.7 (0.7) | 0.82 | 0.70 | 0.87 |
| I17 | 0.6 (0.8) | 1.26 | 0.78 | 0.89 |
| I18 | 1.3 (0.9) | 0.29 | -0.64 | 0.95 |
| I19 | 0.6 (0.8) | 1.00 | 0.15 | 0.92 |
| I20 | 1.0 (0.9) | 0.65 | -0.57 | 0.94 |
| I21 | 0.4 (0.8) | 1.91 | 2.98 | 0.89 |

for the overall model and 0.86 to 0.95 for each variable indicated the data were adequately sampled and appropriate for analysis. Parallel Analysis suggested the number of components to retain was three, while the Minimum Average Partial method achieved a minimum of 0.01 with two components. Scree plot suggested five components having eigenvalues bigger than one. After considering the result from PA, MPA and Scree plots, three components of DASS were retained.

Three components accounting for 48.7% of the total variance were extracted using a combination of PA, MAP, and the scree plot. Table 3 shows the standardised loadings (pattern matrix) of 21 items loaded on the three extracted components. The first component explained 25.2% of the total variance and retained five stress items and two anxiety items with factor loadings bigger than 0.6. For the second and third components, four depression and one stress item were retained. Altogether, four depression items (5, 16, 17, 21), six stress items (1, 6, 8, 11, 12, 14), and two anxiety items (9, 15) were retained to form the abbreviated DASS version with 12 items in this study.

The factor structure and model fit of DASS-21, the current abbreviated 12-item DASS, and five other abbreviated versions were tested. (Table 4). The current DASS-12 showed an acceptable model fit when three-factor were tested ($\chi^2$/df = 4.39, CFI = 0.935, TLI = 0.916, RMSEA = 0.086, SRMR = 0.0447). Compared to the DASS-21, all abbreviated DASS versions demonstrated better CFI and TLI. Among these, the Malaysian DASS-12 was the best fit three-

**Table 3. Standardised factor loading (Wave 1 data, n = 519).**

| Items | RC1* | RC2* | RC3* |
|---|---|---|---|
| Depression | | | |
| I3 | -0.048 | 0.516 | 0.266 |
| I5 | -0.152 | 0.112 | 0.713 |
| I10 | -0.113 | 0.560 | 0.415 |
| I13 | 0.288 | 0.527 | 0.114 |
| I16 | -0.199 | 0.153 | 0.757 |
| I17 | 0.184 | 0.741 | -0.035 |
| I21 | 0.018 | 0.822 | -0.008 |
| Stress | | | |
| I1 | 0.672 | 0.184 | -0.089 |
| I6 | 0.646 | 0.004 | 0.005 |
| I8 | 0.830 | -0.034 | -0.038 |
| I11 | 0.735 | 0.124 | -0.011 |
| I12 | 0.770 | 0.105 | -0.038 |
| I14 | -0.006 | -0.084 | 0.607 |
| I18 | 0.558 | 0.090 | 0.026 |
| Anxiety | | | |
| I2 | 0.072 | -0.085 | 0.411 |
| I4 | 0.310 | -0.020 | 0.220 |
| I7 | 0.518 | -0.289 | 0.246 |
| I9 | 0.694 | 0.041 | 0.061 |
| I15 | 0.874 | -0.035 | -0.177 |
| I19 | 0.387 | -0.257 | 0.359 |
| I20 | 0.510 | 0.380 | -0.167 |
| Variance explained | 25.2% | 13.3% | 10.2% |

*RC1, 2, 3: Extracted component 1, 2, 3

**Table 4. Confirmatory factor analysis results (Wave 2 data, n = 455).**

| DASS versions | Structure | $\chi^2$/df | CFI | TLI | RMESA | SRMR |
|---|---|---|---|---|---|---|
| DASS-8 | Three-factor | 5.09 | 0.956 | 0.928 | 0.095 | 0.047 |
| | Two-factor | 5.10 | 0.951 | 0.927 | 0.095 | 0.047 |
| Korean DASS-12 | Three-factor | 3.27 | 0.935 | 0.915 | 0.071 | 0.043 |
| | Two-factor | 4.57 | 0.893 | 0.867 | 0.089 | 0.057 |
| | Bifactor | 2.07 | 0.975 | 0.960 | 0.049 | 0.033 |
| Malaysian DASS-12 | Three-factor | 2.43 | 0.953 | 0.939 | 0.056 | 0.046 |
| | Two-factor | 3.25 | 0.923 | 0.904 | 0.070 | 0.054 |
| DASS-14 | Three-factor | 2.97 | 0.937 | 0.923 | 0.066 | 0.049 |
| | Two-factor | 3.97 | 0.909 | 0.891 | 0.078 | 0.058 |
| DASS-18 | Three-factor | 4.15 | 0.864 | 0.842 | 0.083 | 0.067 |
| | Two-factor | 4.09 | 0.865 | 0.845 | 0.082 | 0.067 |
| DASS-21 | Three-factor | 4.18 | 0.863 | 0.845 | 0.084 | 0.065 |
| | Two-factor | 4.16 | 0.863 | 0.847 | 0.083 | 0.065 |
| Current DASS-12 | Three-factor | 4.39 | 0.935 | 0.916 | 0.086 | 0.067 |
| | Two-factor | 5.01 | 0.921 | 0.901 | 0.094 | 0.070 |

factor model ($\chi^2$/df = 2.43, CFI = 0.953, TLI = 0.939, RMSEA = 0.056, SRMR = 0.046). The two-factor model was tested for all abbreviated versions. DASS-8 showed the best model fit when the two-factor model was tested ($\chi^2$/df = 5.10, CFI = 0.951, TLI = 0.927, RMSEA = 0.095, SRMR = 0.047).

The bifactor model was also tested for all abbreviated versions, but identification problems occurred when dealing with factors containing two variables. Even after two methods were applied to solve the problem, some models could not be identified, and no solution could be obtained for the bifactor models (S1 Appendix). The only applicable model was the Korean DASS-12, which provided a better fit ($\chi^2$/df = 2.07, CFI = 0.975, TLI = 0.960, RMSEA = 0.049, SRMR = 0.033) (Fig 2) compared to the original three-factor structure. Among all the potential structures, the Korean DASS-12 bifactor model had a better fit than the Malaysian three-factor model and the DASS-8 two-factor model, which was the best structure measuring psychological distress among young adults.

As the Korean DASS-12 was considered the best abbreviated DASS version, known-group, and concurrent (criterion) validity as well as reliability were subsequently evaluated for young adults with and without TMD symptoms and compared with the DASS-21 version using the combined data (n = 864) from wave 1 and wave 2. A total of 110 participants were removed from the combined data as they reported only headaches without other TMD symptoms. The

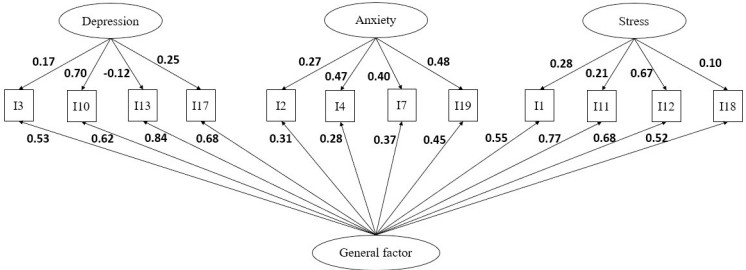

**Fig 2. The Korean DASS-12 bifactor structure and factor loadings (standardised).** Values on the arrows specifies the standardized factor loading correspond to the specific factor and the general factor.

**Table 5. Known-groups validity of the DASS-21 and Korean DASS-12 (Combined data, n = 864).**

| Variables | DASS-21 | | | Korean DASS-12 | | |
|---|---|---|---|---|---|---|
| | 5T-positive | 5T-negative | p-value | 5T-positive | 5T-negative | p-value |
| **Total DASS** | | | <0.001* | | | 0.001* |
| **Mean (SD)** | 18.5 (10.7) | 14.7 (9.8) | | 9.5 (6.0) | 7.3 (5.4) | |
| **Median (IQR)** | 17 (11–25) | 14 (7–21) | | 9 (5–13) | 7 (3–11) | |
| **Depression** | | | 0.003* | | | <0.001* |
| **Mean (SD)** | 4.4 (3.7) | 3.6 (3.5) | | 2.5 (2.4) | 2.0 (2.2) | |
| **Median (IQR)** | 3 (2–7) | 3 (1–5) | | 2 (1–4) | 1 (0–3) | |
| **Anxiety** | | | <0.001* | | | <0.001* |
| **Mean (SD)** | 6.3 (3.8) | 4.7 (3.3) | | 2.6 (2.2) | 1.7 (1.7) | |
| **Median (IQR)** | 6 (3–9) | 4 (2–7) | | 2 (1–4) | 1 (0–3) | |
| **Stress** | | | <0.001* | | | <0.001* |
| **Mean (SD)** | 7.8 (4.4) | 6.4 (4.2) | | 4.4 (2.7) | 3.6 (2.6) | |
| **Median (IQR)** | 7 (5–11) | 6 (3–9) | | 4 (2–6) | 4 (2–5) | |

*Results of Mann-Whitney U test (p < 0.05)

combined data contained 864 participants aged 18 to 25 (females = 79.5% and males = 20.5%) with a mean age of 21.0 (SD = 1.8). Of these, 55.4% (n = 385) were 5Ts-positive and 44.6% (n = 479) were 5Ts-negative.

The normality assumption of the Korean DASS-12 and the DASS-21 scores was examined using the Shapiro-Wilk test. The subscale and total scores of DASS-12 and DASS-21 were not normally distributed (all p <0.001).

A non-parametric Mann-Whitney test was utilised to explore the difference in the Korean DASS-12 and DASS-21 subscales and the total score between young adults with and without TMD symptoms. Significantly higher scores were found in DASS-12 Depression (p = 0.001), Anxiety (p < 0.001), Stress (p < 0.001), and the total score (p < 0.001) in adults with TMD symptoms than those without. Similarly, significantly higher scores were identified in DASS-21 Depression (p = 0.003), Anxiety (p < 0.001), Stress (p < 0.001) and total score (p < 0.001). The Korean DASS-12 showed good known-groups validity when contrasted to the DASS-21 (Table 5).

Correlation analyses were performed using Spearman correlation between DASS-12 and DASS-21 subscales and total scores. The result indicated that the Korean DASS-12 total score had a significant strong positive correlation with the DASS-21 total score ($r_s$ = 0.959, p < 0.001) as well as Depression ($r_s$ = 0.927, p < 0.001), Stress ($r_s$ = 0.941, p < 0.001) and Anxiety ($r_s$ = 0.817, p < 0.001). The Korean DASS-12 thus showed good concurrent (criterion) validity compared to the DASS-21.

The reliability was computed using Cronbach's Alpha coefficients. An adequate reliability of the Korean DASS-12 total score and subscales was obtained (Depression = 0.772, Anxiety = 0.601, Stress = 0.801, Total = 0.861) compared to the DASS-21 (Depression = 0.826, Anxiety = 0.752, Stress = 0.849, Total = 0.918).

## Discussion

This study aimed to shorten the DASS-21 using PCA, confirm the factor structure and model fit of various abbreviated DASS-21 versions, and identify and establish the validity/reliability of the optimal version for assessing young adults with TMDs. PCA was used to determine items to be retained in the shortened version, resulting in three extracted components and 12

items (Depression: 5, 16, 17, 21, Anxiety: 9, 15, Stress: 1, 6, 8, 11, 12, 14). CFA was performed to explore and test the factor structure, and the Korean DASS-12 was found to be the best-fit model. It was considered the best abbreviated DASS-21 version for screening young adults with TMD symptoms for psychological distress. The Korean DASS-12 also showed good known-groups validity, concurrent (criterion) validity, and reliability when contrasted to DASS-21.

Shorter questionnaires are preferred to increase participation and data quality in both clinical and research settings. This is the primary motivation for examining the structure of DASS-21 and attempting to shorten it through various methods. The most commonly used methods were exploratory factor analysis (EFA) and principal component analysis (PCA). In this study, PCA was chosen as the method of analysis instead of EFA. Theoretically, PCA is performed when there is a large set of variables, and the aim is to reduce them to score on composite variables that retain as much information as possible. This study aimed to reduce the number of items in the DASS, and items that contained the most information were identified and formed the current DASS-12. EFA is preferred when the factor structure is unclear, and the aim is to explore the appropriate number of underlying factors that could be extracted from the observed data.

The meaningful threshold set for factor loadings should be both practical and statistically significant [39]. It is common to arbitrarily consider factor loadings of 0.32 or 0.40 as salient (just statistical consideration). However, in this study, the threshold for factor loading was set to 0.6 to achieve item reduction at a practical significance level as well. Only two items were considered insignificant if the factor loading threshold was set at 0.4. Moreover, three items were identified as insignificant if the threshold was set at 0.5. Twelve items were considered significant only when a factor loading of 0.6 was set, forming an abbreviated version with fewer items while retaining as much information as possible.

There were several methods to extract the data in CFA. The most common estimations were maximum likelihood and least square methods. Many researchers adopted the maximum likelihood method because they attempted to generalize to the overall population and compute model parameters [40]. However, this method required normally distributed data. Others recommended least square methods because they do not have distribution assumptions and are sensitive to weak factors (factors with weak correlations). However, large sample sizes will be required [41]. Comparing the two methods, a study found that maximum likelihood methods generally have smaller CFI than least square methods (unweighted least squares (ULS) or diagonally weighted least squares (DWLS)) because when using Diagonally Weighted Least Squares (DWLS), the influence of threshold distribution on population CFI was found to be minimal [42]. This study selected maximum likelihood because our data did not deviate from normality severely, and the correlations between Depression, Anxiety, and Stress were strong based on previous studies [1].

Seven abbreviated DASS-21 versions were evaluated together with three different factor structures (two-factor, three-factor, and bifactor structures). Model fit varied when comparing three-factor and two-factor structures. The three-factor structure consistently demonstrated better model fit than the two-factor structure in all versions. (Table 4). The bifactor structure was applied where feasible, and the Korean DASS-12 bifactor structure showed better model fit than the three-factor and two-factor structures and was the best-fit model among all versions and structures appraised. Though Ali [14] demonstrated that the DASS-8 had a good factor structure and adequate psychometrics, an identification problem existed and could not be resolved when fitting a bifactor model to it. Compared to the Korean DASS-12, DASS-8 contains three Depression items, three Anxiety items, and only two Stress items. The Korean version of the DASS-12 is more balanced as each scale comprises four items. While a shorter

questionnaire may be preferable, reducing the number of questions would result in less information gathered. The need to be comprehensive and parsimonious must, therefore, be balanced.

The current DASS-12 showed an acceptable model fit in CFA. Theoretically, it should be the best model based on the data. However, the Korean DASS-12 demonstrated better-fit indexes than the current DASS-12. One possible reason was how the items were retained. Osman [12] suggested that "four potential items might more clearly delineate each dimension" based on previous studies. For each scale, they retained four items based on the factor analysis results. This study reduced the number of items by the significance of factor loading, and the number of items retained in each subscale was not equal (four Depression items, six Stress items, and two Anxiety items). Further validation work would be recommended.

DASS was recommended to utilize the total score with strong evidence for validity with young adults in one recent systematic review [43]. For depression and anxiety subscale, the relationship was found to vary. This finding followed the conclusion, that in the Korean DASS-12, the total score had the highest coefficients ($\alpha = 0.861$).

When testing validity and reliability, participants with headaches alone were excluded when combining wave 1 and 2 data. Although the contemporary DC/TMD standard has "headaches attributed to TMDs" as a diagnostic subtype and patients with painful TMDs were more likely to have headaches [44], primary headaches such as migraine, tension-type, and cluster headaches, are widespread affecting about 46% of the general population, and can be caused by many other diseases or conditions [45, 46]. Individuals with just headaches and no other TMD symptoms were thus omitted to enhance precision.

The Korean DASS-12 was regarded as the best abbreviated DASS-21 version for screening young adults, which has been previously validated in both Korean population and Polish adults [13, 47]. Compared to the DASS-21, this short version appears to have an acceptable factorial structure, as found in these two validation studies. The Korean study also tested and achieved satisfactory results in content, convergent, discriminant, concurrent, and known-groups validity, as well as internal consistency, indicating its potential use in clinical and research settings. However, the sensitivity and specificity of the Korean DASS-12 version require further evaluation by comparing it with the Diagnostic and Statistical Manual of Mental Disorders [48], which remains the "gold standard" for diagnosing mental disorders.

There were a few limitations in this study. First, the gender distribution was unequal, with females comprising most of the sample. Due to the higher likelihood of females experiencing TMD than males [20], more female students may be interested in participating in the study during recruitment, resulting in the study participants being predominantly females. There could be a risk of bias due to the gender imbalance. Future studies with equal gender distribution would be preferred to confirm the results. Second, only concurrent (criterion) validity, known-group validity, and reliability were evaluated between the DASS-21 and the Korean DASS-12 in this study. Convergent validity with other scales measuring psychological distress was not evaluated and should be included in future research endeavors. Population norms for the Korean DASS-12 values are also required to establish the cut-points for three subscale severity ratings. Additionally, this study's findings must be confirmed in other age groups and countries.

## Conclusion

A shorter 12-item version of the DASS-21 was derived using PCA. The current DASS-12 showed an acceptable model fit in the three-factor structure. However, comparing the seven abbreviated DASS-21 versions, the Korean DASS-12 possessed the best model fit. It was

considered the best abbreviated DASS-21 version for screening young adults with TMD symptoms for psychological distress. The Korean DASS-12 presented good known-group validity for the three subscales and the overall measure, as well as good concurrent (criterion) validity and reliability when contrasted to the DASS-21. As the Korean DASS-12 demonstrated better psychometric performance, it is recommended for research and clinical use. Further validation studies using different populations are needed to verify the measurement properties of the Korean DASS-12 in different age groups as well as ethnicities.

## Supporting information

**S1 Appendix.**
(DOCX)

**S1 Dataset.**
(XLSX)

## Author Contributions

**Conceptualization:** Yinghao Xiong, Adrian Ujin Yap, Carolina Marpaung, May Chun Mei Wong.

**Data curation:** Carolina Marpaung.

**Formal analysis:** Yinghao Xiong.

**Methodology:** Yinghao Xiong, Adrian Ujin Yap, Carolina Marpaung, May Chun Mei Wong.

**Supervision:** Adrian Ujin Yap, Carolina Marpaung, May Chun Mei Wong.

**Visualization:** Yinghao Xiong.

**Writing – original draft:** Yinghao Xiong.

**Writing – review & editing:** Adrian Ujin Yap, Carolina Marpaung, May Chun Mei Wong.

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
