## [Decision Letter · Decision Letter 0]

2 Oct 2024

PONE-D-24-35163The Depression, Anxiety, Stress Scales-21: Principal Component Analysis and Evaluation of Abbreviated Versions in Young Adults with Temporomandibular DisordersPLOS ONE

Dear Dr. Wong,

Thank you for submitting your manuscript to PLOS ONE. After careful consideration, we feel that it has merit but does not fully meet PLOS ONE’s publication criteria as it currently stands. Therefore, we invite you to submit a revised version of the manuscript that addresses the points raised during the review process.

**ACADEMIC EDITOR: **

**Please follow the recommendations of the reviewers. In addition, please include recommendations for further validation of this simplified instrument, based on criterion validity and optimisation of the instrument (sensitivity or specificity). Furthermore, in the flowcharts, please specify what the values or coefficients placed on the arrows of the structural diagrams correspond to. This can be done at the bottom of the figure. **

**Include in the discussion the advantages and disadvantages of applying principal component analysis vs. parallel analysis, or whether it is advantageous to include both.**

We look forward to receiving your revised manuscript.

Kind regards,

César Félix Cayo-Rojas, Ph.D.

Academic Editor

PLOS ONE

**Journal Requirements:**

3. In the online submission form, you indicated that The datasets used and/or analysed during the current study are available from the corresponding author on reasonable request.

**Additional Editor Comments:**

Please follow the recommendations of the reviewers. In addition, please include recommendations for further validation of this simplified instrument, based on criterion validity and optimisation of the instrument (sensitivity or specificity). Furthermore, in the flowcharts, please specify what the values or coefficients placed on the arrows of the structural diagrams correspond to. This can be done at the bottom of the figure. 

Reviewers' comments:

Reviewer's Responses to Questions

**Comments to the Author**

1. Is the manuscript technically sound, and do the data support the conclusions?

Reviewer #1: Yes

Reviewer #2: Yes

2. Has the statistical analysis been performed appropriately and rigorously? 

Reviewer #1: Yes

Reviewer #2: Yes

3. Have the authors made all data underlying the findings in their manuscript fully available?

Reviewer #1: Yes

Reviewer #2: Yes

4. Is the manuscript presented in an intelligible fashion and written in standard English?

Reviewer #1: Yes

Reviewer #2: Yes

5. Review Comments to the Author

**Reviewer #1:** 1. Citations and references such as “Lee et al., 2019” should not be included in the manuscript abstract, specifically in 14 spacing. It is suggested that any external references be removed to ensure that the abstract is self-sufficient and fully understandable without recourse to the full text.

2. In the discussion, paragraph 8, line spacing 6, there has been an error in the automatic referencing of the manuscript, reflected in the message “Error! Reference source not found”. It is necessary to review the referencing system and verify that all citations are properly linked to their respective sources in the bibliography or reference list.

3. Check bibliographic references

1,3,4,5,6,7,8,9,10,12,13,14,15,16,17,19,20,21,22,24,25,26,29-43 y 45

Please note the attached Example taken from instructions to the author from Plos One journal:

Devaraju P, Gulati R, Antony PT, Mithun CB, Negi VS. Susceptibility to SLE in South Indian Tamils may be influenced by genetic selection pressure on TLR2 and TLR9 genes. Mol Immunol. 2014 Nov 22. pii: S0161-5890(14)00313-7. doi: 10.1016/j.molimm.2014.11.005.

Note:...When providing a DOI, follow the format of the example above with the label and full DOI included at the end of the reference (doi: 10.1016/j.molimm.2014.11.005). Do not provide an abbreviated DOI or URL.

**Reviewer #2:** Thank you for inviting to review the manuscript: The Depression, Anxiety, Stress Scales-21: Principal Component Analysis and Evaluation of Abbreviated Versions in Young Adults with Temporomandibular Disorders.

In relation to the document read, I consider the great effort made by the authors in proposing this paper could improve the following:

- Discussion: The discussion could be more exploited by comparing with other studies done in this regard.

6. PLOS authors have the option to publish the peer review history of their article (what does this mean?). If published, this will include your full peer review and any attached files.

Reviewer #1: **Yes: **Huamani-Echaccaya Jose Luis

Reviewer #2: No

---

## [Author Response · Author response to Decision Letter 0]

6 Nov 2024

We have addressed all the comments from the reviewers.

---

## [Decision Letter · Decision Letter 1]

16 Dec 2024

The Depression, Anxiety, Stress Scales-21: Principal Component Analysis and Evaluation of Abbreviated Versions in Young Adults with Temporomandibular Disorders

PONE-D-24-35163R1

Dear Dr. May Chun Mei Wong,

We’re pleased to inform you that your manuscript has been judged scientifically suitable for publication and will be formally accepted for publication once it meets all outstanding technical requirements.

Kind regards,

César Félix Cayo-Rojas, Ph.D.

Academic Editor

PLOS ONE

Additional Editor Comments (optional):

Dear Author, I am pleased to announce that your manuscript has been accepted for publication. Thank you for responding to all the comments, the reviewers and I have been satisfied with your responses.

Reviewers' comments:

Reviewer's Responses to Questions

**Comments to the Author**

1. If the authors have adequately addressed your comments raised in a previous round of review and you feel that this manuscript is now acceptable for publication, you may indicate that here to bypass the “Comments to the Author” section, enter your conflict of interest statement in the “Confidential to Editor” section, and submit your "Accept" recommendation.

Reviewer #1: All comments have been addressed

Reviewer #2: All comments have been addressed

2. Is the manuscript technically sound, and do the data support the conclusions?

Reviewer #1: Yes

Reviewer #2: Yes

3. Has the statistical analysis been performed appropriately and rigorously? 

Reviewer #1: Yes

Reviewer #2: Yes

4. Have the authors made all data underlying the findings in their manuscript fully available?

Reviewer #1: Yes

Reviewer #2: Yes

5. Is the manuscript presented in an intelligible fashion and written in standard English?

Reviewer #1: Yes

Reviewer #2: Yes

6. Review Comments to the Author

Reviewer #1: I am grateful to the authors for satisfactorily addressing the suggestions made. The manuscript reflects a marked improvement and is ready to be considered for the next stage.

Reviewer #2: Good work, despite its limitations. Good contribution to the area of knowledge.There is an adequate statistical treatment.

7. PLOS authors have the option to publish the peer review history of their article (what does this mean?). If published, this will include your full peer review and any attached files.

Reviewer #1: **Yes: **Huamani-Echaccaya JL

Reviewer #2: No

---

## [Editor Report · Acceptance letter]

15 Jan 2025

PONE-D-24-35163R1 

PLOS ONE

Dear Dr. Wong, 

I'm pleased to inform you that your manuscript has been deemed suitable for publication in PLOS ONE. Congratulations! Your manuscript is now being handed over to our production team.

Kind regards, 

on behalf of

Dr. César Félix Cayo-Rojas 

Academic Editor

PLOS ONE